# Uncoupling of the Astrocyte Syncytium Differentially Affects AQP4 Isoforms

**DOI:** 10.3390/cells9020382

**Published:** 2020-02-07

**Authors:** Shirin Katoozi, Nadia Skauli, Soulmaz Zahl, Tushar Deshpande, Pascal Ezan, Claudia Palazzo, Christian Steinhäuser, Antonio Frigeri, Martine Cohen-Salmon, Ole Petter Ottersen, Mahmood Amiry-Moghaddam

**Affiliations:** 1Division of Anatomy, Department of Molecular Medicine, Institute of Basic Medical Sciences, University of Oslo; 0315 Oslo, Norway; shirin.katoozi@medisin.uio.no (S.K.); nadia.skauli@medisin.uio.no (N.S.); soulmaz.rahmani@medisin.uio.no (S.Z.); ole.petter.ottersen@ki.se (O.P.O.); 2Institute of Cellular Neuroscience, Medical Faculty, University of Bonn, Venusberg-Campus 1, 53127 Bonn, Germany; tushard1987@gmail.com (T.D.); Christian.Steinhaeuser@ukbonn.de (C.S.); 3Physiology and Physiopathology of the Gliovascular Unit Research Group. Center for Interdisciplinary Research in Biology (CIRB), College de France, Unité Mixte de Recherche 7241 CNRS, Unité1050 INSERM, PSL Research University, 75005 Paris, France; pascal.ezan@college-de-france.fr (P.E.); martine.cohen-salmon@college-de-france.fr (M.C.-S.); 4Department of Basic Medical Sciences, Neurosciences and Sense Organs, School of Medicine, University of Bari Aldo Moro, 70124 Bari, Italy; clapalazzo@hotmail.it (C.P.); antonio.frigeri@uniba.it (A.F.)

**Keywords:** AQP4, astrocytes, Cx43, Cx30, gap junction, polarization

## Abstract

The water channel protein aquaporin-4 (AQP4) and the gap junction forming proteins connexin-43 (Cx43) and connexin-30 (Cx30) are astrocytic proteins critically involved in brain water and ion homeostasis. While AQP4 is mainly involved in water flux across the astrocytic endfeet membranes, astrocytic gap junctions provide syncytial coupling allowing intercellular exchange of water, ions, and other molecules. We have previously shown that mice with targeted deletion of *Aqp4* display enhanced gap junctional coupling between astrocytes. Here, we investigate whether uncoupling of the astrocytic syncytium by deletion of the astrocytic connexins Cx43 and Cx30 affects AQP4 membrane localization and expression. By using quantitative immunogold cytochemistry, we show that deletion of astrocytic connexins leads to a substantial reduction of perivascular AQP4, concomitant with a down-regulation of total AQP4 protein and mRNA. Isoform expression analysis shows that while the level of the predominant AQP4 M23 isoform is reduced in Cx43/Cx30 double deficient hippocampal astrocytes, the levels of M1, and the alternative translation AQP4ex isoform protein levels are increased. These findings reveal a complex interdependence between AQP4 and connexins, which are both significantly involved in homeostatic functions and astrogliopathologies.

## 1. Introduction

Astrogliopathology is now emerging as a feature common to several neurological conditions [1,2]. In particular, loss or mislocalization of astrocytic transporter or channel molecules has been associated with Alzheimer’s disease, stroke, and various forms of human and experimentally induced epilepsy [3,4,5,6]. The prevailing concept is that astrocytes have important homeostatic functions that critically depend on the specific localization of membrane molecules.

Gap junctions and Aquaporin-4 (AQP4) water channels are prominently involved in water and ion homeostasis in the brain. Connexin-43 (Cx43, *Gja1*) and connexin-30 (Cx30, *Gjb6*) are the major connexins of astrocytes, with regional differences in their expression. In the hippocampus, astrocytic gap junctions are mainly composed of Cx43 while in other regions such as the thalamus, Cx30 is the prevailing connexin [7,8,9]. Gap junctions assemble into intramembranous plaques which couple adjacent astrocytes into a syncytium allowing spatial redistribution of water ions and larger molecules up to a size of 1 kD [10,11]. Moreover, several lines of evidence suggest that connexins interact with the cell cytoskeleton and are involved in migration, proliferation and cell adhesion [12]. AQP4, on the other hand, does not couple neighboring astrocytes but is instead concentrated in the astrocytic plasma membrane domain that abuts on brain capillaries [13]. AQP4 is anchored to the perivascular astrocytic membrane domains through an interaction with α-syntrophin, a PDZ domain containing member of the dystrophin-associated protein complex (DAPC), which links the actin cytoskeleton to the basal lamina proteins agrin and laminin [14,15,16,17,18]. Based on its localization, AQP4 would be expected to regulate the volume of the astrocytic syncytium at large. This is in line with several studies which implicate the perivascular AQP4 pool in the buildup and resolution of brain edema [15,19,20,21,22,23]. The two main isoforms of AQP4, M1 (the longer isoform) and M23 (the shorter and more abundant isoform) differ in their ability to form orthogonal arrays of particles (OAPs) and are thought to be involved in adhesion and cell migration in addition to water transport [24,25]. A newly characterized isoform, AQP4ex, is generated by translational readthrough and contains a 29 amino acid *C*-terminal extension, which is believed to be involved in proper membrane localization of AQP4 and interaction with other intracellular proteins [26,27].

Little is known about the factors that control the expression of connexins and AQP4 in brain astrocytes. However, evidence is emerging that connexins and AQP4 might be mutually interdependent in terms of their expression and membrane assembly. Targeted deletion of AQP4 was found to facilitate dye coupling between hippocampal astrocytes [28]. This is consistent with our finding of an increased number of astrocytic gap junctions in AQP4 KO mice [29]. Conversely, a significant loss of total AQP4 protein was observed in mice with a double knockout of the astrocytic gap junction proteins Cx43 and Cx30 [30].

Here we explore the interdependence between astrocytic connexins and aquaporins—two classes of molecule critically involved in homeostatic functions in brain—by assessing whether an uncoupling of the astrocyte syncytium through double knockout of Cx30 and Cx43 affects membrane specific and isoform specific expression of AQP4. In particular, using quantitative immunogold cytochemistry, we ask whether uncoupling of the astrocytic syncytium affects the astrocytic endfoot pool of AQP4 and hence the overall capacity for water transport between astrocytes and the perivascular space.

## 2. Materials and Methods

### 2.1. Animals

C57/BL6 mice (Jackson Laboratories, Boulder, CO) and Cx30−/−; Cx43 fl/fl:hGFAP-Cre (Cx43/30 double knockout (dKO) mice bred on a C57/BL6 background were used in this study. To generate the dKO animals, mice lacking Cx43 in astrocytes were obtained by interbreeding of Cx43fl mice [31] with mice carrying the hGFAP-cre transgene [32]. Subsequently, Cx43fl/fl:GFAP-Cre mice were crossed with Cx30−/− mice [33] to receive mice lacking Cx30 and Cx43 in astrocytes. The dKO mice were then backcrossed to a C57/BL6 background. All animals had *ad libitum* access to food and drinking water. All animal experiments were performed according to the European Council Law on Protection of Laboratory Animals, and were approved by The Norwegian Animal Research Authority (NARA), the French Agency for Animal Experimentation and Animal Ethics Committee (Université Paris Descartes, agreement no. 86 to 23), and Italian 189/2017-PR and n° 2A298.N.2G1).

### 2.2. Perfusion and Tissue Preparation for Electron Microscopy

Brain sections of adult (3 months old) male wild type (WT) and Cx43/30 dKO mice (*n* = 4 for each genotype) were prepared after anesthesia and transcardial perfusion with 4% formaldehyde in 0.1 M phosphate buffer (PB) at pH 6.0, then pH 10.0 using pH shift protocol without addition of picric acid as previously described [34]. After perfusion, the brains were removed and post-fixed overnight in the fixation solution and stored in 1:10 dilution of the same fixative in 0.1 M PB.

### 2.3. Postembedding Immunogold Electron Microscopy

Brain sections were harvested and cut into 0.5–1 mm tissue blocks. Hippocampus and parietal cortex were dissected, cryo-protected and quick-frozen in liquid propane (−170 °C) and subjected to freeze substitution. Specimens were embedded in methacrylate resin (Lowicryl HM20) and polymerized by UV light below 0 °C [35]. 80 nm ultrathin sections from parietal cortex and hippocampus were cut using an ultrotome (Reichert Ultracut S, Leica) and placed on 300mesh grids.

Immunogold cytochemistry was performed as previously described [34,36]. Briefly, ultrathin sections were incubated overnight with primary antibodies (Table 1) diluted in Tris-buffered saline with 0.1% Triton X-100 (TBS-T) with 2% (*w/v*) human serum albumin (HSA) at room temperature (RT). Thereafter, the sections were rinsed with TBS-T (3 × 10 min) and incubated for 2 h in TBS-T containing 2% (*w/v*) HSA (10 min) before incubation with goat anti-rabbit (GAR) secondary antibody coupled to colloidal 15 nm gold particles (Abcam, Cambridge, UK), diluted in TBS-T containing 2% (*w/v*) HSA and polyethylene glycol (PEG) (1:20).

Finally, the sections were rinsed with dH_2_O (3 × 10 min), dried and incubated in 2% uranyl acetate (90 s) followed by an incubation in 0.3% lead citrate (90 s). Immunolabelled sections were examined in a TECNAI 12 transmission electron microscope at 60 kV (FEI, Hillsboro, OR, USA).

### 2.4. Immunogold Quantitation and Data Analysis

Quantitative analysis was performed as previously described [36,37,38]. Briefly, to assess the expression of AQP4 and α-syntrophin in the perivascular astrocytic endfeet in parietal cortex and hippocampus, images of 20–30 capillaries were acquired randomly from each section. To check whether the possible changes in the AQP4 perivascular pool is due to mislocalization, around 40 images were also taken of neuropil without capillaries from each section. All the images were taken at 26500× magnification. The linear densities of gold particles were determined by using analysis software (Soft Imaging Systems (SIS), Münster, Germany). Investigators were blinded regarding animal genotypes during processing and analysis. Linear densities were analyzed semi-automatically and the data was transferred to SPSS Version 22 (SPSS, Chicago, IL, USA) for statistical analysis. Sample groups were statistically compared using ANOVA with Bonferroni’s post hoc test. Data are presented as mean SEM. *p <* 0.05 was considered to be significant.

### 2.5. Preparation of Total Protein Lysates from Brain Regions

Mice were subjected to euthanasia in a CO_2_ chamber. Brains were isolated and kept on ice cold petri dishes. Samples were briefly rinsed with PBS. Hippocampi and cortices were dissected quickly. They were snap frozen in liquid nitrogen and stored at −80 °C. Cortex and hippocampus from 8 week old WT (*n* = 3) and dKO mice (*n* = 3) were homogenized in RIPA buffer (50 mM Tris-HCl pH 7.4; 150mM NaCl; 5mM EDTA; 1% Triton X-100; 0.5% sodium deoxycholate; 0.1% SDS), with freshly added 1× SigmaFAST protease inhibitor cocktail (Sigma-Aldrich, St. Louis, MO, USA) and 1× PhosSTOP phosphatase inhibitor (Roche Life Science, Basel, Switzerland). Homogenates were prepared by mechanical dissociation using lysing matrix tubes (MP Biomedicals) and incubated on ice for 30 min before centrifugation at 14000 rpm at 4 °C for 15 min. The supernatant was collected as total protein and concentrations were measured using a Pierce™ BCA protein assay kit (Thermo Fisher, Waltham, MA, USA).

### 2.6. SDS-PAGE and Western Blotting

Samples were heated in 1× Laemmli sample buffer at 37 °C for 10 min and 10 μg samples were separated on 4–20% Criterion™ 18-well gels (BioRad, Hercules, CA, USA) by SDS-PAGE using the Criterion™ (BioRad) Tris-glycine system at 185V for 1 h 15min at 4 °C. Proteins were transferred to 0.2 µm Immun-Blot polyvinylidene fluoride (PVDF) membranes (BioRad) by wet blotting at 100 V for 30 min at 4 °C. Uniform transfer of proteins was verified by reversible Ponceau S (0.1% *w/v*, 1% acetic acid, Sigma-Aldrich) staining.

Membranes were blocked for 1 h at RT in 5% BSA in 1X Tris-buffered saline (TBS) (BioRad) and washed 3 × 5 min in 1X TBS with 1% Tween-20 (Sigma-Aldrich). Membranes were cut before separate overnight incubation with primary antibodies (Table 1) at 4 °C. Subsequently, membranes were washed in TBS-T before incubation with secondary HRP-conjugated antibodies (Table 1) for 1 h at RT. Membranes were washed 3 × 10 min in TBS-T before detection of immunoreactive bands by SuperSignal™ West Pico Chemiluminescent Substrate (Thermo Fisher) using a Fujifilm LAS-3000 (Fujifilm, Tokyo, Japan) and ChemiDoc™ Touch (BioRad) imaging system. For these two systems, secondary HRP-conjugated antibody concentrations were diluted 1:5000 and 1:25,000, respectively. The AQP4 M1 isoform antibody was generated by GeneScript towards the predicted antigen MSDRAAARRWGKC within the rat M1-AQP4 specific sequence.

Bands were quantified as arbitrary background-subtracted density units in Image Studio Lite (Ver 5.2, Licor Biosciences, Lincoln, NE, USA). Normalization was performed by dividing intensities of protein bands of interest with the normalizing control α-tubulin band intensity for their respective lane. The obtained values were transferred to SPSS Version 25 (SPSS, Chicago, IL, USA) and compared using the non-parametric Kruskal–Wallis test. Data are presented as percentages of the average control with medians (95% CI). *p <* 0.05 was considered to be significant.

### 2.7. RNA Isolation and Reverse Transcriptase Quantitative PCR (RT-qPCR)

Total RNA was isolated from cortex and hippocampus using the RNeasy Plus Mini Kit (QIAGEN). The RNA concentration and integrity were determined using a NanoDrop 2000c spectrophotometer (Thermo Scientific) and agarose gel electrophoresis. cDNA was synthesized using 400 ng of RNA from each sample. The reaction was performed by GoScript Reverse Transcription System (Promega) using Oligo (dT)_15_. All the cDNA samples were diluted in 10 mM Tris-HCl (pH 8.0) to 2.5 ng/μL. Copy numbers of *Aqp4* were calculated using absolute standards. The qPCR was performed in a total volume of 20 μL, containing the Power SYBR Green PCR Master Mix (Applied Biosystems), forward and reverse primers (10 µM), and 2 μL of the cDNA template. Thermal cycling was performed on the StepOnePlus system (Applied Biosystems) with the following conditions: 95 °C for 10 min, followed by 40 cycles at 95 °C for 15 s and 60 °C for 1 min. Standard curve, No Reverse Transcriptase (NRT) and No Template Control (NTC) were included in the study. Primers used are listed in Table 2. *Gapdh*, *Tbp*, *Hprt1* and *Actb* were evaluated for use as internal controls by Norm Finder [39] and a combination of *Gapdh* and *Hprt1* weas selected for double normalization as they showed the best stability value of 0.060. Independent samples *t*-test was used for data analysis. Data are presented as mean and SEM. *p < 0.05* was considered to be significant.

## 3. Results

### 3.1. Deletion of Astroglial Connexins Leads to a Significant Decrease in Perivascular AQP4 and Abolishes the Regional Heterogeneity in AQP4 Distribution

Quantitative high resolution immunogold cytochemistry using an antibody against AQP4 revealed a decrease in the density of perivascular AQP4 immunogold particles in parietal cortex and hippocampus of Cx43/30 dKO mice compared to WT (Figure 1A–D). In agreement with our previous studies, quantitative analysis of immunogold labeling in WT mice showed about 38% higher AQP4 immunogold linear density in the hippocampus (mean 16.45) compared to parietal cortex (mean 11.93, Figure 1E, *p* < 0.001). This regional difference was abolished in Cx43/30 dKO mice as the reduction in the perivascular AQP4 immunogold density was more pronounced in hippocampus (34%) than in parietal cortex (20%) (Figure 1E).

To resolve if the decrease in the perivascular AQP4 immunogold density is due to a redistribution of AQP4, we quantified the AQP4 immunogold labeling in the neuropil outside the perivascular zone. Our analysis showed that deletion of astroglial connexins was associated with a significant decrease in AQP4 immunogold density also in the neuropil, both in hippocampus and parietal cortex (Figure 1F).

### 3.2. Reduction in the Perivascular Pool of AQP4 is Independent of α-Syntrophin

To assess whether the reduction observed in the perivascular pool of AQP4 in Cx43/30 dKO mice is due to down-regulation of the AQP4 anchoring molecule, α-syntrophin, we compared the perivascular density of this molecule in both genotypes. Our data did not show any significant differences in the perivascular density of α-syntrophin between the two groups (Figure 2).

### 3.3. Deletion of Astroglial Connexins Leads to a Decrease in Aqp4 Transcript, but Has a Differential Effect on AQP4 Isoforms at the Protein Level

RT-qPCR analysis demonstrated a significant decrease in *Aqp4* transcripts in the parietal cortex and hippocampus of Cx43/Cx30 dKO mice (Figure 3A). Similarly, Western blots provided evidence for a reduction in the total amounts of total AQP4 protein and the main AQP4 isoform (M23) in the Cx43/Cx30 dKO cortex and hippocampus compared to WT controls (Figure 3B). These changes were significant for both total AQP4 and AQP4 M23 in both regions (Figure 3C,D). A band corresponding to the AQP4 M1 isoform was identified (Figure 3B arrow) but this was not distinct enough to allow for quantitative analysis.

To determine whether the remaining AQP4 isoforms M1 and AQP4-ex were altered in the Cx43/30 dKO animals, we used specific antibodies towards these isoforms. While total AQP4 and the AQP4-M23 isoform showed decreased expression in dKO mice, we observed an up-regulation of the AQP4-M1 and AQP4-ex isoforms in Cx43/30 dKO animals compared to WT (Figure 4A).

The increase in AQP4-M1 was modest yet significant (Figure 4B). The AQP4-ex protein level showed a 7-fold increase when comparing the medians of dKO and WT mice (Figure 4C).

## 4. Discussion

Astrocytic gap junctions and AQP4 are complementary in terms of their localization and function. While gap junctions are conduits for redistribution of water and other molecules within the astrocytic syncytium, AQP4 serves as an influx and efflux route for water between individual astrocytes and the perivascular space. The question explored in the present paper is whether deletion of connexins–entailing uncoupling of the astrocytic syncytium—affects the expression of specific AQP4 isoforms in specialized membrane domains. Our data provide new insight in how AQP4 expression is regulated and have implications for our understanding of brain pathophysiology as altered expression and function of astroglial connexins is seen in several neurological conditions [40,41,42,43]. The present results indicate that dysregulation of connexins in the context of neurological diseases might be associated with loss or mislocalization of AQP4 water channels.

We have previously shown that there is a regional heterogeneity in the perivascular AQP4 density.Specifically, the perivascular AQP4 density in hippocampus is about 20% higher than the perivascular AQP4 density in parietal cortex [36]. In Cx43/Cx30 dKO animals, the decrease in perivascular AQP4 was more pronounced in hippocampus than in parietal cortex, thus abolishing the regional heterogeneity. Interestingly, previous studies have shown that the size of astrocytic syncytium, assessed by gap junction mediated dye coupling, is larger in hippocampus than in neocortex [44]. Given that clearance of metabolically produced water to the perivascular space is one of the main functions of the perivascular AQP4 channels [23], one can speculate that size of astrocytic syncytium regulates the density of the perivascular AQP4, leading to a higher density of perivascular AQP4 in hippocampus than in neocortex. This might explain why the regional heterogeneity in the perivascular AQP4 density is abolished in dKO mice lacking a functional astrocyte syncytium.

AQP4 M23 is the most predominant AQP4 isoform in brain. This is the isoform that constitutes the orthogonal arrays of proteins (OAPs) that accumulate in astrocytic endfeet around brain microvessels. Here we show that the perivascular expression of this isoform is reduced in Cx43/30 dKO mice, as is the total amount of AQP4 in the brain. The latter finding is consistent with that of Ezan, Andre, Cisternino, Saubamea, Boulay, Doutremer, Thomas, Quenech’du, Giaume and Cohen-Salmon [30]. The reduction of perivascular AQP4 observed in our study, was more pronounced in hippocampus than in parietal cortex. Our data indicate that Cx43/30 dKO mice have a reduced capacity for water transport across astrocytic plasma membranes and specifically between astrocytes and the perivascular space in the hippocampus. In line with this, Lutz, et al. [45] noted swelling of astrocytes in the hippocampus of mice subjected to deletion of Cx43 and Cx30, although in the latter study deletion of astrocytic connexins was under control of the mouse GFAP promoter.

The mechanisms that underlie down-regulation of M23 after Cx43/30 dKO are not known. AQP4 expression is under the influence of several micro-RNAs [46] and further studies are required to determine whether these are involved in the alterations seen in the present study.

As for the astrocytic connexins, the M1 isoform of AQP4 is known to play a role in astrocytic migration. Expression of this isoform reduces the size of OAPs, which predominantly contain the M23 isoform [24]. The up-regulation of M1 after Cx43/30 dKO could therefore be a compensation for the non-channel functions of the connexins, such as migration [24] or process motility [47]. This is particularly relevant for astrogliosis, in which AQP4 and Cx43 might play similar roles [48].

Among the AQP4 isoforms investigated AQP4ex was the one showing the most conspicuous change in dKO mice. The level of this isoform—which constitutes about 10% of the total AQP4 pool—was almost seven-fold higher after connexin deletion. AQP4ex is the longest of the AQP4 isoforms and is formed by translational readthrough. With its long tail, the AQP4ex isoform might play a specific role in anchoring OAPs to the dystrophin complex in astrocytic endfeet [26]. If the up-regulation of AQP4ex translates into a more efficient anchoring of perivascular M23 it would serve to constrain the reduction of the perivascular AQP4 pool secondary to the down-regulation of M23 expression. The long C-terminus of AQP4-ex might also substitute for the C-terminus of Cx43 in interaction with intracellular proteins. This idea will be followed up in future studies.

## 5. Conclusions

We previously demonstrated that targeted deletion of *Aqp4* leads to an increased number of gap junctions in astrocytic endfoot membrane domains [29] corresponding to an enhanced functional coupling [28]. This increase occurs at the post-translational level as the amounts of Cx43 and Cx30 transcripts and protein were unchanged [29]. Here we show that connexins and AQP4 are mutually interdependent as deletion of Cx43/Cx30 causes a down-regulation of AQP4 M23 and a sizeable up-regulation of the newly discovered AQP4ex. The down-regulation of M23 AQP4 (the predominant AQP4 isoform in brain) is paralleled by a reduced level of *Aqp4* mRNA indicating regulation at the transcriptional level. Thus, while astrocytic connexins and AQP4 are mutually interdependent, there is a striking asymmetry when it comes to the mechanisms involved. The present data unravel a complex interaction between Cx43, Cx30, and AQP4 that might be an important feature in astrogliopathology.

## Figures and Tables

**Figure 1 cells-09-00382-f001:**
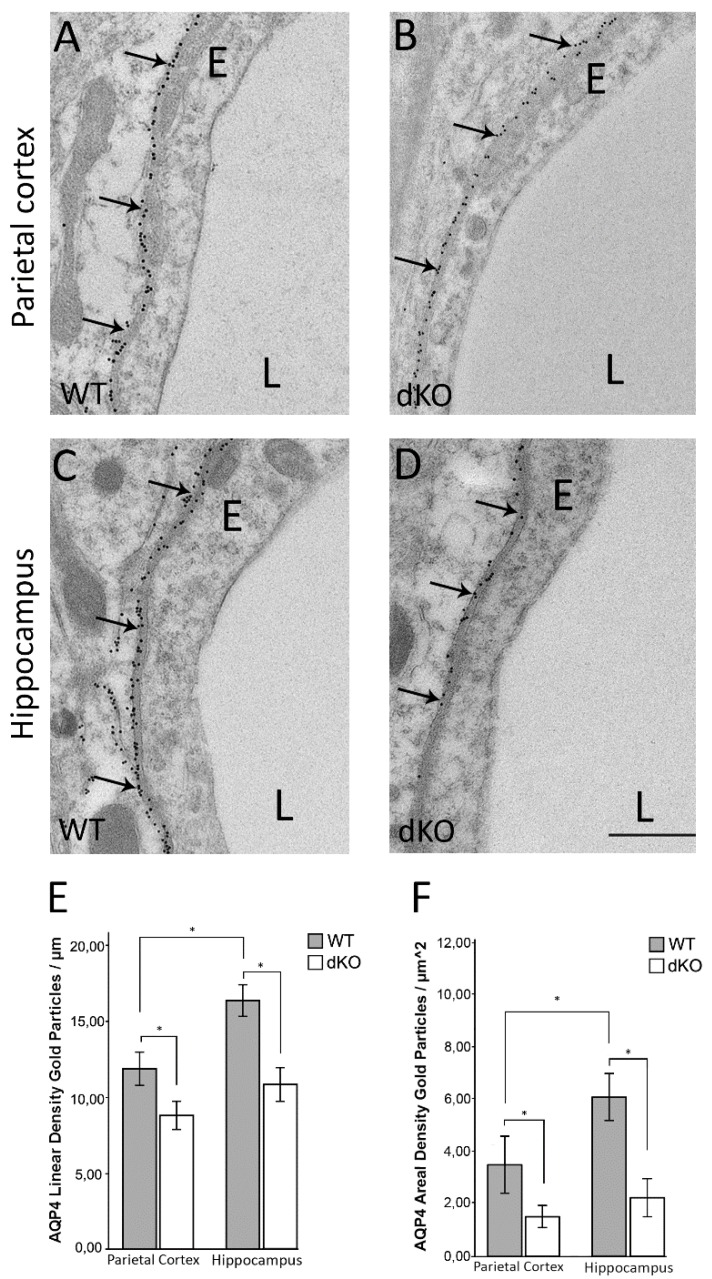
Electron micrographs of parietal cortex and hippocampus of WT and Cx43/30 dKO mice showing AQP4 immunogold labeling. (**A**–**D**) The AQP4 immunogold labeling is concentrated in the perivascular membrane domain facing endothelium (arrows) **(E)** Quantitative analysis of perivascular AQP4 immunogold labeling in parietal cortex and hippocampus of Cx43/30 dKO and WT mice. The linear density of gold particles was significantly lower than in WT controls. (*p* < 0.001). (**F**) Quantitative analysis of AQP4 immunogold labeling in randomly selected micrographs of neuropil outside the perivascular zone in parietal cortex and hippocampus of Cx43/30 dKO and WT mice. AQP4 labeling intensity was significantly lower in both parietal cortex and hippocampus of Cx43/30 dKO compared to WT (*p* < 0.005). E; endothelial cells, L; vessel lumen. Scale bar: 500 nm. * significant difference according to ANOVA with Bonferroni’s post hoc test; *n* = 5 for each genotype; *error bars* indicate SEM. *p* < 0.05.

**Figure 2 cells-09-00382-f002:**
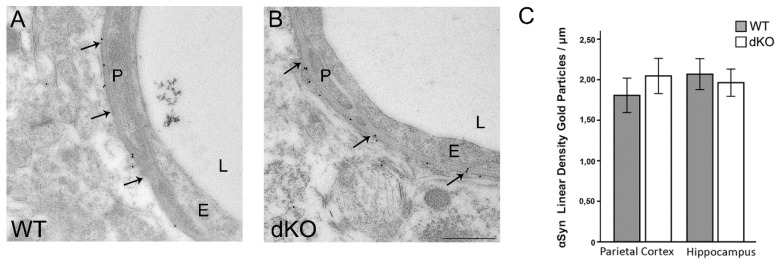
Immunogold analysis of perivascular α-syntrophin expression in WT and Cx43/30 dKO mice. (**A**,**B**) Electron micrographs of hippocampus from WT and dKO mice shows that α-syntrophin immunogold labeling is concentrated in the perivascular membrane domain (arrows). (**C**) Quantitative analysis of α-syntrophin immunogold labeling in parietal cortex and hippocampus of Cx43/30 dKO and WT. No significant difference was observed between the genotypes. E; endothelial cells, P; pericyte, L; vessel lumen. Scale bar: 500 nm. * significant difference according to ANOVA with Bonferroni’s post hoc test, *n* = 5 for each genotype; *error bars* indicate SEM. *p* < 0.05.

**Figure 3 cells-09-00382-f003:**
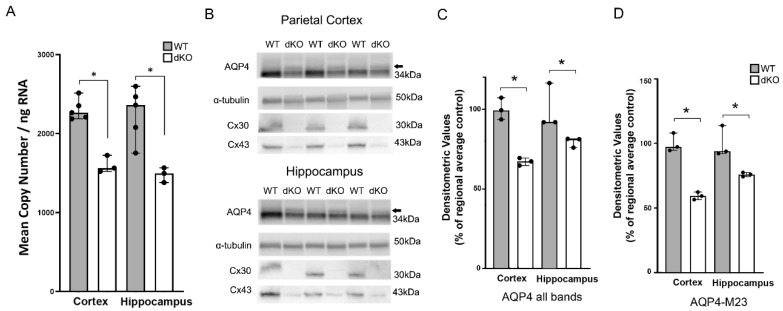
RT-qPCR and Western blot analysis of AQP4 in WT and Cx43/30 dKO mice. (**A**) RT-qPCR analysis of parietal cortex and hippocampus in WT and Cx43/30 dKO mice. Graph illustrates quantification of *Aqp4* mRNA. Significant decrease in the mRNA level of *Aqp4* was found in the samples of *Gja1* and *Gjb6* dKO compared to WT mice in each region. A combination of *Gapdh* and *Hprt1* was used for normalization of copy numbers. * Significant difference according to independent samples *t-*test; *n* = 3 for each genotype; *error bars* SEM; *p* < 0.05. (**B**) Representative immunoblots of total protein lysates from hippocampus and parietal cortex for AQP4 in WT and Cx43/30 dKO animals. Two major bands detected in homogenates from both regions correspond to M1 (arrow) and M23 isoforms of AQP4. α-tubulin was used as the loading control. dKO animals are negative for Cx30 and Cx43 protein. (**C**) Quantitation of total AQP4 in cortex and hippocampus of WT and Cx43/30 dKO mice. Significant decrease in total AQP4 protein was found in both regions of Cx43/30 dKO animals compared to WT mice. (**D**) Quantitation of the AQP4 M23 isoform in cortex and hippocampus of WT and Cx43/30 dKO mice. Significant decrease in the AQP4 M23 was found in both regions of Cx43/30 dKO animals compared to WT. Values are presented as average of the respective wild type regional control. Individual values are presented as black dots. * Significant difference according to non-parametric Kruskal–Wallis test; *n* = 3 for each genotype; *error bars* indicate medians with 95% CI, *p* < 0.05.

**Figure 4 cells-09-00382-f004:**
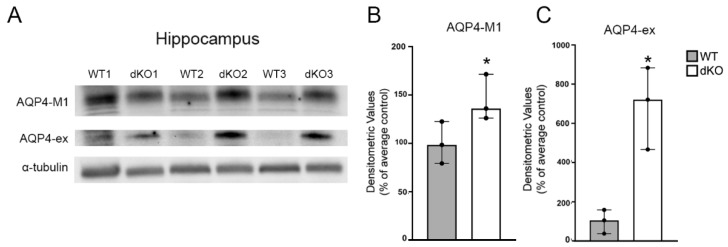
Immunoblots and quantification of total protein lysates from hippocampus of WT and Cx43/30 dKO mice. (**A**) Representative immunoblots of total protein lysates from hippocampus for AQP4-M1 and AQP4-ex in WT and Cx43/30 dKO animals. α-tubulin was used as the loading control. (**B**,**C**) Graphs illustrate densitometric analysis of AQP4-M1 and AQP4-ex immunoblotts of hippocampus of WT and Cx43/30 dKO mice. A significant increase in AQP4-M1 and AQP4-ex protein isoforms was found in Cx43/30 dKO animals compared to WT. Values are presented as average of the respective wild type regional control *significant difference with non-parametric Kruskal–Wallis test; *n* = 3 for each genotype; *error bars* medians with 95% CI, *p* < 0.05.

**Table 1 cells-09-00382-t001:** Antibodies. The following antibodies used in this study.

Methods	Primary Antibody	Secondary Antibody
Immunogold	Rabbit anti-AQP4, 1:500, Sigma-Aldrich, A5971	Goat anti-rabbit 15 nm, 1:20, Abcam
	Rabbit anti-Cx30, 1:200, Invitrogen, 71-2200	Goat anti-rabbit 15 nm, 1:20, Abcam
	Rabbit anti-Cx43, 1:200, Invitrogen, 71-0700	Goat anti-rabbit 1 nm, 1:20, Abcam
	Rabbit anti-α-syntrophin (SYN259), 1:200, Gift from Dr Marvin E. Adams	Goat anti-rabbit 1 nm, 1:20, Abcam
Western Blotting	Rabbit anti-AQP4, 1:2000, Sigma-Aldrich, A5971	Donkey anti-rabbit HRP, 1:5000, Amersham, GE Life Sciences
	Rabbit anti-Cx43, 1:1000, Sigma-Aldrich, C6219	Donkey anti-rabbit HRP, 1:5000, Amersham, GE Life Sciences
	Rabbit anti-Cx30, 1:200, Invitrogen, 71-2200	Donkey anti-rabbit HRP, 1:5000, Amersham, GE Life Sciences
	Rabbit anti-α-tubulin, 1:2000, Abcam, ab4074	Donkey anti-rabbit HRP, 1:5000 and 1:25,000, Amersham, GE Life Sciences
	Custom Rabbit polyclonal anti-mouse AQP4-ex antibody, 1:2000, GeneScript [26]	Donkey anti-rabbit HRP, 1:25,000, Amersham, GE Life Sciences
	Custom Rabbit polyclonal anti-AQP4-M1 antibody, 1:2000, GeneScript	Donkey anti-rabbit HRP, 1:25,000, Amersham, GE Life Sciences

**Table 2 cells-09-00382-t002:** PCR primers. The following primer pairs used in the study.

Gene	Forward Primer	Reverse Primer
*Aqp4*	5′-TTTGGACCCGCAGTTATCAT-3′	5′-GTTGTCCTCCACCTCCATGT-3′
*Gja1*	5′-GTGCCGGCTTCACTTTCATTAAG-3′	5′-AAATGAAGAGCACCGACAGC-3′
*Gjb6*	5′-GACATTCCCACTGTGACCCT-3′	5′-TCGTGCAGGCTTATTCTGAGT-3′
*Gapdh*	5′-TGCGACTTCAACAGCAACTC-3′	5′-CTTGCTCAGTGTCCTTGCTG-3′
*Hprt1*	5′-GCCCCAAAATGGTTAAGGTT-3′	5′-TTGCGCTCATCTTAGGCTTT-3′

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
