# Peer review of "Uncoupling of the Astrocyte Syncytium Differentially Affects AQP4 Isoforms"

_cells, 2020, doi:10.3390/cells9020382_

Round 1

Reviewer 1 Report

This study provided an important  information for us, and these data may be clue to understand a feature of astroglial pathology.

Minor points:

Line 83: Authors should write details of GFAPcre mouse. ex. mouse strain.

Line 107: What % of tritonX-100 in TBS.

Fig.1 A-D, Fig.2 A-B: Although these pictures are excellent, it is hard to see  the gold particles due to small. Authors should show these pictures more larger.

Fig.1E-F, Fig. 3C-D, Fig.4B-C: The character size of axis labels is too small. These should be bigger and clearer. 

Author Response

We thank the reviewer for constructive comments. Below, you find a point-by-point response to all the comments.

Reviewer 1

“This study provided an important  information for us, and these data may be clue to understand a feature of astroglial pathology.

RESPONSE: We thank the reviewer for this comment.

Minor points:

Line 83: Authors should write details of GFAPcre mouse. ex. mouse strain.”

RESPONSE: We have now expanded the animal parts of the methods explaining in more detail how the mice were generated, providing more appropriate references.

Line 107: What % of tritonX-100 in TBS.

RESPONSE: 0.1%- added now to the text.

Fig.1 A-D,
Fig.2 A-B:
Although these pictures are excellent, it is hard to see  the gold particles due to small.
Authors should show these pictures more larger.”

RESPONSE: Thank you for your comment. We realized that the EM images in figure 1 are at a lower magnification than those in figure 2.  We have now enlarged the EM images in Fig. 1A-D, including arrows that indicate the perivascular astrocytic membranes where the gold particles are localized. Hopefully, this will make the gold particles easier to see.
Fig 2 A-B is already at high magnification, and we can unfortunately not increase the size of the image without losing structural information. However, the figure is supplied at 300dpi, and the reader should be able to enlarge it substantially and see the details with high resolution.  

“Fig.1E-F,
Fig. 3C-D,
Fig.4B-C: The character size of axis labels is too small. These should be bigger and clearer.“

Thank you for this observation. We have now increased the size of the axis labels and text that seemed too small in Fig.1E-F, Fig. 3C-D and Fig.4B-C.

Reviewer 2 Report

The authors investigated the relationship between connexins (Cx43, Cx30) and AQP4 and revealed that deletion of Cxs reduced total AQP4 of perivascular area but the isoform expressions were differently regulated: M23 form was decreased whileAQP4ex were increased, respectively.  This is the significant study to investigate the Cx’s role for aquaporin in detail, however, some points should be revised or supplemented with appropriate explanations.

Cx43 and Cx30 were focused among twenties of subtypes of Cxs due to the majority occupation in astrocytes? Is there an interaction region with Cx43 in AQP4ex isoform? In introduction, the authors described there was a report that AQP4 deletion facilitated gap junction functions. It seemed to be natural for cells that GJ compensates for the transfer of intercellular waters or the other ions. However, the finding of this study was the opposite direction, Cx depletions introduced total AQP4 downregulation. Please add some more discussions or speculations in the view points of physiological meanings.

Author Response

We thank the reviewer for constructive comments. Below, you find a point-by-point response to all the comments.

 This is the significant study to investigate the Cx’s role for aquaporin in detail, however, some points should be revised or supplemented with appropriate explanations.

RESPONSE: We thank the reviewer and have now expanded the discussions sections to explain the findings better.

 “Cx43 and Cx30 were focused among twenties of subtypes of Cxs due to the majority occupation in astrocytes?”

RESPONSE: Indeed, Cx43 and Cx30 are the major connexins of astrocytes. We have now phrased this more clearly in the text.

“Is there an interaction region with Cx43 in AQP4ex isoform?”

RESPONSE: This is currently unknown, but we are currently following up this exciting topic in a new project.

“In introduction, the authors described there was a report that AQP4 deletion facilitated gap junction functions. It seemed to be natural for cells that GJ compensates for the transfer of intercellular waters or the other ions. However, the finding of this study was the opposite direction, Cx depletions introduced total AQP4 downregulation.”

RESPONSE: Thank you for bringing up this question. We have now added a paragraph to the discussions where we try to speculate on why we see this difference.

 “Please add some more discussions or speculations in the view points of physiological meanings. “

RESPONSE: As mentioned above, we have expanded the text in the discussions section trying to explain and speculate) on what these findings mean. However, we will follow this question in our ongoing and future studies.